# Resistance Monitoring for Six Insecticides in Vegetable Field-Collected Populations of *Spodoptera litura* from China

**Ziyi Zhang** [1,2,†], **Bingli Gao** [2,†], **Cheng Qu** [2], **Jingyu Gong** [1], **Wenxiang Li** [1,*], **Chen Luo** [2] and **Ran Wang** [2,*]

1   College of Agriculture and Forestry Technology, Hebei North University, Zhangjiakou 075000, China; zhangziyi130523@163.com (Z.Z.); gjyu2021@163.com (J.G.)

2   Institute of Plant Protection, Beijing Academy of Agriculture and Forestry Sciences, Beijing 100097, China; gaobinglishine@163.com (B.G.); qucheng@ipepbaafs.cn (C.Q.); luochen@ipepbaafs.cn (C.L.)

\*   Correspondence: liwenxiang9769@163.com (W.L.); wangran@ipepbaafs.cn (R.W.)

†   These authors contributed equally to this work.

**Abstract:** The common cutworm, *Spodoptera litura* (Fabricius), is a notorious and damaging insect pest of horticultural crops in China, the management of which largely relies on chemical agents that are limited by the development of chemical resistance in target populations. As such, resistance monitoring of *S. litura* populations is a necessary part of management strategies of insecticide resistance. In the current work, we monitored resistance to six insecticides in field-collected populations of *S. litura* sampled from eleven provinces across China in 2021. The results show that *S. litura* populations developed significant resistance against chlorantraniliprole, cyantraniliprole, metaflumizone, and pyridalyl and low levels of resistance to chromafenozide. However, *S. litura* populations were susceptible or exhibited minimal resistance to tetraniliprole. Possible cross-resistances between chlorantraniliprole, cyantraniliprole, metaflumizone, pyridalyl, and chromafenozide were found by pairwise correlation, which also revealed that tetraniliprole lacked cross-resistance with all insecticides tested. Our results suggest suspending the use of chemical agents against which *S. litura* displayed significant field-evolved resistance, such as chlorantraniliprole, metaflumizone, and pyridalyl, in favor of pesticides against which *S. litura* was susceptible or exhibited minimal resistance, such as tetraniliprole and chromafenozide, which may help slow the development of insecticide resistance, and in which field management programs aimed at controlling *S. litura* in China would benefit from the integration of such survey-informed insecticide application strategies. Moreover, the baseline susceptibility confirmed for the six tested insecticides can contribute to design strategies of resistance management for *S. litura*.

**Keywords:** *Spodoptera litura*; sensitivity of insecticide; field-evolved resistance; cross-resistance; management of resistance

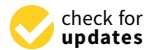



## 1. Introduction

The tobacco cutworm, *Spodoptera litura* (Fabricius), is a notorious insect pest of horticultural crops around the world that causes significant damage to a wide range of crops, including tobacco, cotton, soybeans, and vegetables [1]. In part due to its high reproductive capacity, over-reliance on chemical agents to control *S. litura* has caused resistance to a variety of chemical agents used worldwide [2]. The first report of pesticide resistance in *S. litura* populations was against benzene hexachloride, described in 1965 [3]. In recent years, more and more field-collected populations of *S. litura* have evolved high-level resistance to different types of chemical agents, such as organophosphates, pyrethroids, carbamates, and even several newer chemistries including abamectin, indoxacarb, emamectin benzoate, chlorantraniliprole, metaflumizone and cyantraniliprole [4–12]. In the above cases, mechanisms of resistance to chlorantraniliprole and metaflumizone could be associated with changed activities of detoxification enzymes and mutations of the target gene, respectively [7,12].

One approach to slowing evolution of resistance involves rotating newer pesticides that have distinct modes of action with existing insecticides.

In recent years, tetraniliprole, chromafenozide, and pyridalyl, three novel chemical agents, were reported to provide good control against lepidopteran pests even at low dosages [13–15]. Owing to their efficacy and convenience in the field, insecticides are heavily used in controlling insect pests; however, overuse of insecticides has given rise to the significant development of insecticide resistance in pest populations, thereby reducing the efficacy of currently and widely used pesticide chemistry. As described previously in many other species of lepidopteran pests, gradual selection pressure resulting from continual and long-term insecticide application in the field has greatly contributed to the development of resistance [16–20]. To date, few reports of resistance against the aforementioned three novel chemical agents in *S. litura* populations have been recorded in China.

In our current work, we monitored the status of resistance to six insecticides (chlorantraniliprole, metaflumizone, pyridalyl, cyantraniliprole, chromafenozide, and tetraniliprole) in field-collected populations of *S. litura* from eleven provinces of China in the year 2021. Moreover, pairwise correlation analysis revealed patterns of cross-resistance to all the tested pesticides in eleven field-collected populations of *S. litura*. Our results provide valuable data concerning the resistance level of chemical agents in *S. litura* populations in China and suggestions for sustainable strategies of resistance management.

## 2. Materials and Methods

### 2.1. Insects

The reference strain of *S. litura*, which was reared in a chamber with no exposure to any chemical agents over five years, provided by the Henan bio company, was used as the susceptible Lab-S strain. As shown in Figure 1 and Table 1, eleven populations of *S. litura* were individually sampled in 2021 from eleven provinces of China: Hubei (WH), Anhui (HF), Jiangxi (NC), Jiangsu (YC), Zhejiang (LS), Hunan (CS), Fujian (ND), Guangdong (GZ), Hainan (SY), Guangxi (GL), and Yunnan (YX). About 200 fourth-instar larvae of *S. litura* were sampled randomly in different host plants (Table 1) and were maintained in a rearing room to obtain F$_1$ progeny for the bioassays. In the above work, an artificial diet and 10% sugar solution were used to maintain the tested populations of *S. litura* [7]. All tested *S. litura* were reared under well-controlled conditions of relative humidity as 60–70%, temperature as 26 ± 1 °C, and a light/dark photoperiod as 16:8 h.

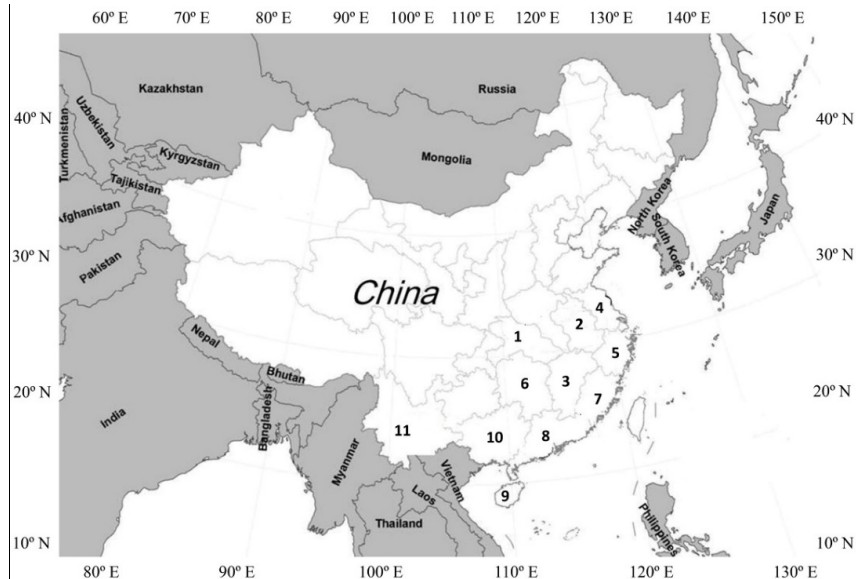

**Figure 1.** Field sites in China from which *S. litura* samples were collected; each number in the map represents each site.

**Table 1.** Detailed information on collected *S. litura* field populations.

| Population | Map Ref. No. | Location of Collection | Site | Date and Host Plant |
|---|---|---|---|---|
| WH | 1 | Wuhan, Hubei, central China | 41.19 N, 123.11 E | July 2021, lotus root |
| HF | 2 | Hefei, Anhui, eastern China | 41.59 N, 120.50 E | July 2021, lotus root |
| NC | 3 | Nanchang, Jiangxi, central China | 39.97 N, 116.31 E | August 2021, lotus root |
| YC | 4 | Yancheng, Jiangsu, eastern China | 39.73 N, 116.69 E | September 2021, lotus root |
| LS | 5 | Lishui, Zhejiang, eastern China | 39.35 N, 117.10 E | September 2021, lotus root |
| CS | 6 | Changsha, Hunan, central China | 38.90 N, 116.94 E | September 2021, lotus root |
| ND | 7 | Ningde, Fujian, eastern China | 40.58 N, 115.00 E | July 2021, taro |
| GZ | 8 | Guangzhou, Guangdong, southern China | 38.82 N, 115.39 E | July 2021, lotus root |
| SY | 9 | Sanya, Hainan, southern China | 34.91 N, 113.56 E | July 2021, pepper |
| GL | 10 | Guilin, Guangxi, southern China | 34.33 N, 113.75 E | July 2021, taro |
| YX | 11 | Yuxi, Yunnan, southern China | 36.78 N, 117.23 E | August 2021, lotus root |

*2.2. Insecticides and Chemicals*

The insecticides utilized were analytically standardized (Table 2). Chlorantraniliprole (Dr. Ehrenstorfer, CAS# 500008-45-7, catalog# DRE-C11145000), tetraniliprole (Dr. Ehrenstorfer, CAS# 1229654-66-3, catalog# DRE-C17414700) and chromafenozide (Dr. Ehrenstorfer, CAS# 143807-66-3, catalog# DRE-C11665500) were purchased from Dr. Ehrenstorfer, Germany. Metaflumizone (Sigma Aldrich, Shanghai, China, CAS# 139968-49-3, catalog# 32966-100MG), cyantraniliprole (Sigma Aldrich, CAS# 736994-63-1, catalog# 32372-25MG), pyridalyl (Sigma Aldrich, CAS# 179101-81-6, 32393-25MG), dimethyl sulfoxide (Sigma Aldrich, CAS# 67-68-5, catalog# D8418-500ML) and Triton X-100 (Sigma Aldrich, CAS# 9002-93-1, catalog# 93443-100ML) were purchased from Sigma Aldrich, Shanghai, China.

**Table 2.** Insecticides tested against field-collected *S. litura* populations.

| Insecticide | IRAC Mode of Action Class |
|---|---|
| Metaflumizone | 22B: Voltage-dependent sodium channel blockers |
| Chlorantraniliprole | 28: Ryanodine receptor modulators |
| Cyantraniliprole | 28: Ryanodine receptor modulators |
| Tetraniliprole | 28: Ryanodine receptor modulators |
| Chromafenozide | 18: Ecdysone receptor agonists |
| Pyridalyl | Unknown |

*2.3. Bioassays*

Leaf-dip bioassays were performed according to published methods with minor revision [7]. Third-instar larvae of *S. litura* were collected at random, five serial working concentrations of chemical agent were diluted by the use of sterilized water with 0.1% Triton X-100, and four replicates were set up for each of the working concentrations. The tested leaf disc (5 cm diameter) from the cabbage plant *Brassica oleracea* was immersed in each specific working concentration for 20 s, was dried in the rearing chamber, and then placed into each petri dish (5.5 cm diameter). Ten tested larvae were introduced onto the leaf disc used as each replicate for the treatment, and four replicates were performed for each treatment. All tested *S. litura* were reared under the same controlled conditions in a rearing room.

*2.4. Statistical Analysis*

Responses of concentration mortality, slope values, median lethal concentrations (LC$_{50}$) and their 95% fiducial limits (FLs) were calculated by the use of the software POLO Plus [21]. Between the lab-S strain and each field-collected population, the values of LC$_{50}$ were identified as markedly different if overlap was not observed between the 95% FLs. The resistance ratio (RR) was evaluated as LC$_{50}$ (field-collected population)/LC$_{50}$ (Lab-S), and levels of pesticide resistance is reported by the published method [7]: susceptibility (RR < 5), low level of resistance (RR = 5–10), moderate level of resistance (RR = 10–40), high level of resistance (RR = 40–160), and extremely high level of resistance (RR > 160). Pairwise

correlation coefficients were evaluated among the values of log $LC_{50}$ in field-collected populations and the tested chemical agents by the use of analysis of Pearson's correlation using the software of SPSS [22] to assess cross-resistance among diverse chemical agents.

## 3. Results

### 3.1. Baseline Susceptibility of S. litura-Susceptible Lab-S Strain to Six Insecticides

Currently in China, baseline susceptibilities of *S. litura* to tetraniliprole and chromafenozide have not been assessed, and the baseline values for resistance to chlorantraniliprole, metaflumizone, pyridalyl and cyantraniliprole were determined with regional research. In the current work, the above six chemical agents were selected for establishing baseline susceptibilities of *S. litura* populations from eleven provinces of China, and they determined the susceptibility baseline of the susceptible Lab-S strain as reference (Table 3).

**Table 3.** Baseline susceptibility of *S. litura* to six insecticides in the susceptible strain Lab-S.

| Insecticide | N [a] | $LC_{50}$ (95% CI; mg/L) [b] | Slope $\pm$ SE | $X^2$ (df) | *p* Value |
|---|---|---|---|---|---|
| Metaflumizone | 200 | 4.264 (3.379–5.259) | 2.250 $\pm$ 0.287 | 1.003 (3) | 0.79 |
| Chlorantraniliprole | 200 | 2.906 (2.200–3.930) | 1.579 $\pm$ 0.240 | 1.408 (3) | 0.71 |
| Cyantraniliprole | 200 | 1.704 (1.309–2.149) | 1.967 $\pm$ 0.265 | 1.181 (3) | 0.77 |
| Tetraniliprole | 200 | 0.124 (0.100–0.154) | 2.224 $\pm$ 0.276 | 1.557 (3) | 0.70 |
| Chromafenozide | 200 | 1.080 (0.890–1.304) | 2.642 $\pm$ 0.310 | 0.550 (3) | 0.91 |
| Pyridalyl | 200 | 1.394 (1.139–1.672) | 2.844 $\pm$ 0.360 | 0.752 (3) | 0.87 |

[a] Number of tested larvae. [b] Median lethal concentration and 95% confidence interval.

### 3.2. Monitoring Sensitivity to Six Insecticides in Central China

As shown in Figure 2 and Table 4, the three field-collected populations from central China, Wuhan (WH), Changsha (CS) and Nanchang (NC) were sensitive or displayed low levels of resistance to five of the six chemical agents. The WH and CS populations displayed moderate resistance to metaflumizone, at 14.6- and 21.8-fold greater than the Lab-S strain, respectively. Although the NC population (9.6-fold) was classified as having low level resistance to metaflumizone, the resistance level closely approached the moderate range.

**Table 4.** Insecticide susceptibility of *S. litura* collected from central China.

| Population | Insecticide | N [a] | $LC_{50}$ (95% CI; mg/L) [b] | Slope $\pm$ SE | $X^2$ (df) | RR [c] | *p* Value |
|---|---|---|---|---|---|---|---|
| WH | Metaflumizone | 200 | 62.330 (49.193–80.598) | 1.911 $\pm$ 0.260 | 2.283 (3) | 14.6 | 0.51 |
| | Chlorantraniliprole | 200 | 3.431 (2.769–4.350) | 2.176 $\pm$ 0.276 | 2.203(3) | 1.2 | 0.52 |
| | Cyantraniliprole | 200 | 0.991(0.730–1.307) | 1.585 $\pm$ 0.240 | 2.420 (3) | 0.6 | 0.49 |
| | Tetraniliprole | 200 | 0.191 (0.147–0.242) | 1.921 $\pm$ 0.260 | 1.175 (3) | 1.5 | 0.77 |
| | Chromafenozide | 200 | 1.159 (0.937–1.428) | 2.284 $\pm$ 0.282 | 1.785 (3) | 1.1 | 0.68 |
| | Pyridalyl | 200 | 1.103 (0.907–1.333) | 2.602 $\pm$ 0.309 | 2.237 (3) | 0.8 | 0.51 |
| CS | Metaflumizone | 200 | 93.041 (69.574–137.290) | 1.545 $\pm$ 0.246 | 1.980 (3) | 21.8 | 0.65 |
| | Chlorantraniliprole | 200 | 5.827 (4.582–7.534) | 1.881 $\pm$ 0.255 | 2.650 (3) | 2.0 | 0.42 |
| | Cyantraniliprole | 200 | 3.636 (2.766–5.184) | 1.652 $\pm$ 0.252 | 1.627 (3) | 2.1 | 0.69 |
| | Tetraniliprole | 200 | 0.552 (0.393–0.793) | 1.289 $\pm$ 0.229 | 1.462 (3) | 4.5 | 0.71 |
| | Chromafenozide | 200 | 8.643 (6.676–11.041) | 1.847 $\pm$ 0.254 | 0.777 (3) | 8.0 | 0.87 |
| | Pyridalyl | 200 | 8.637 (6.953–10.673) | 2.244 $\pm$ 0.279 | 2.851 (3) | 6.2 | 0.41 |
| NC | Metaflumizone | 200 | 40.901 (32.577–53.488) | 2.042 $\pm$ 0.274 | 1.140 (3) | 9.6 | 0.77 |
| | Chlorantraniliprole | 200 | 1.482 (1.177–1.887) | 1.994 $\pm$ 0.262 | 2.437 (3) | 0.5 | 0.49 |
| | Cyantraniliprole | 200 | 0.819 (0.625–1.034) | 1.960 $\pm$ 0.267 | 1.331 (3) | 0.5 | 0.73 |
| | Tetraniliprole | 200 | 0.893 (0.625–1.106) | 1.338 $\pm$ 0.263 | 2.238 (3) | 7.2 | 0.52 |
| | Chromafenozide | 200 | 1.554 (1.157–2.155) | 1.477 $\pm$ 0.237 | 1.952 (3) | 1.4 | 0.65 |
| | Pyridalyl | 200 | 2.098 (1.614–2.689) | 1.818 $\pm$ 0.253 | 1.789 (3) | 1.5 | 0.67 |

[a] Number of tested larvae. [b] Median lethal concentration and 95% confidence interval. [c] RR: Resistance Ratio = $LC_{50}$ (field-collected population)/$LC_{50}$ (Lab-S).

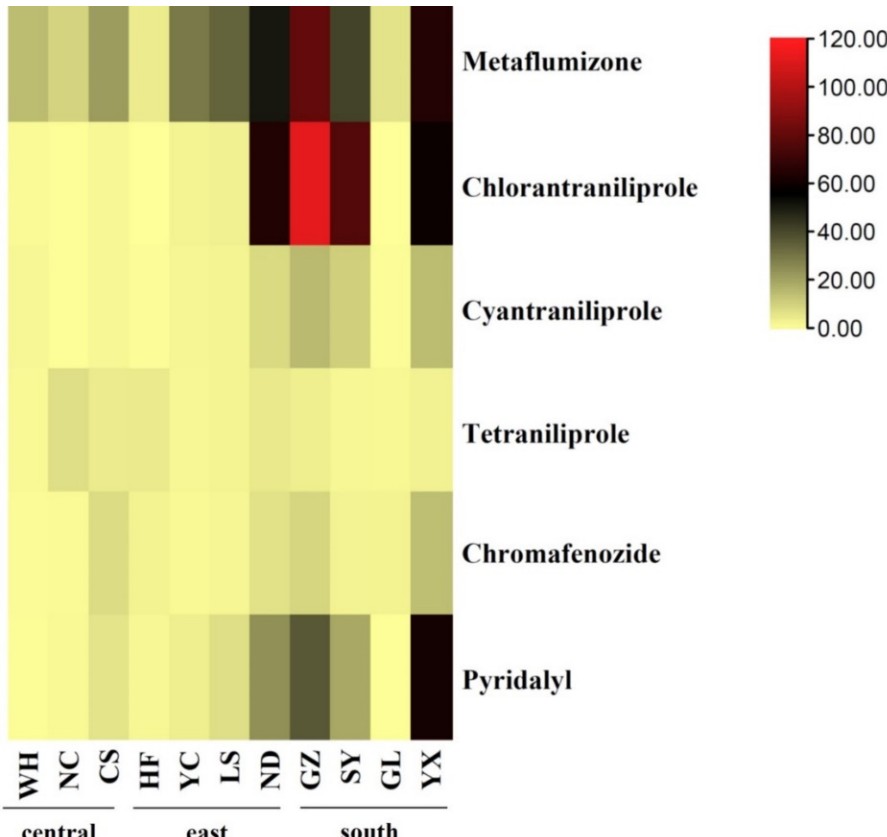

**Figure 2.** Comparative heatmaps of resistance ratios (RR) against six insecticides in eleven field-collected populations of *S. litura* from different regions of China, including central China, east China and south China.

### 3.3. Monitoring Sensitivity to Six Insecticides in Eastern China

Relative resistance levels varied among the four field-collected populations from eastern China (Figure 2 and Table 5). The Hefei (HF) population was susceptible to all six insecticides. In the Yancheng (YC) and Lishui (LS) populations, similar to the three field-collected populations from central China, moderate resistance to metaflumizone (29.3- and 34.1-fold) and low resistance or susceptibility to the other five tested insecticides were observed. Unexpectedly, the Ningde (ND) *S. litura* population displayed high-level resistance to metaflumizone (50.8-fold) and chlorantraniliprole (64.3-fold), and moderate resistance to pyridalyl (24.2-fold). Moreover, the ND population displayed low-level resistance to cyantraniliprole, tetraniliprole, and chromafenozide.

As shown in Figure 2 and Table 6, the Yuxi (YX) population displayed middle- to high-level resistance to five of the six chemical agents, and only showed susceptibility to tetraniliprole. Likewise, Guangzhou (GZ) and Sanya (SY) populations displayed middle- to high-level resistance to four of the six chemical agents, only showing susceptibility or low-level resistance to chromafenozide and pyridalyl. Similar to the ND population, high-level resistance to metaflumizone and chlorantraniliprole were detected in the YX (64.7- and 58.9-fold), GZ (80.2- and 111.6-fold), and SY (41.4- and 76.3-fold) populations, respectively.

**Table 5.** Insecticide susceptibility of *S. litura* collected from eastern China.

| Population | Insecticide | N [a] | LC$_{50}$ (95%CI; mg/L) [b] | Slope $\pm$ SE | X$^2$ (df) | RR [c] | *p* Value |
|---|---|---|---|---|---|---|---|
| HF | Metaflumizone | 200 | 19.373 (14.992–24.558) | 1.912 $\pm$ 0.259 | 2.185 (3) | 4.5 | 0.54 |
| | Chlorantraniliprole | 200 | 1.215 (0.953–1.549) | 1.908 $\pm$ 0.257 | 1.628 (3) | 0.4 | 0.69 |
| | Cyantraniliprole | 200 | 1.119 (0.917–1.358) | 2.529 $\pm$ 0.301 | 1.684 (3) | 0.7 | 0.68 |
| | Tetraniliprole | 200 | 0.582 (0.475–0.711) | 2.425 $\pm$ 0.293 | 2.052 (3) | 4.7 | 0.59 |
| | Chromafenozide | 200 | 3.239 (2.618–4.063) | 2.203 $\pm$ 0.278 | 1.697 (3) | 3.0 | 0.68 |
| | Pyridalyl | 200 | 2.919 (2.351–3.660) | 2.159 $\pm$ 0.273 | 1.549 (3) | 2.1 | 0.70 |
| YC | Metaflumizone | 200 | 124.814 (90.922–179.187) | 1.357 $\pm$ 0.232 | 1.785 (3) | 29.3 | 0.67 |
| | Chlorantraniliprole | 200 | 7.697 (6.028–9.648) | 2.032 $\pm$ 0.266 | 0.578 (3) | 2.6 | 0.90 |
| | Cyantraniliprole | 200 | 3.681 (2.714–5.367) | 1.409 $\pm$ 0.236 | 1.424 (3) | 2.2 | 0.71 |
| | Tetraniliprole | 200 | 0.244 (0.189–0.302) | 2.352 $\pm$ 0.320 | 1.241 (3) | 2.0 | 0.74 |
| | Chromafenozide | 200 | 1.689 (1.296–2.133) | 1.950 $\pm$ 0.297 | 1.980 (3) | 1.6 | 0.65 |
| | Pyridalyl | 200 | 5.262 (4.135–6.746) | 1.897 $\pm$ 0.255 | 1.938 (3) | 3.8 | 0.66 |
| LS | Metaflumizone | 200 | 145.567 (108.926–194.263) | 1.566 $\pm$ 0.238 | 2.166 (3) | 34.1 | 0.54 |
| | Chlorantraniliprole | 200 | 9.474 (6.795–13.104) | 1.372 $\pm$ 0.254 | 2.728 (3) | 3.3 | 0.43 |
| | Cyantraniliprole | 200 | 4.663 (3.602–6.007) | 1.799 $\pm$ 0.250 | 1.293 (3) | 2.7 | 0.76 |
| | Tetraniliprole | 200 | 0.329 (0.176–0.472) | 1.615 $\pm$ 0.294 | 1.972 (3) | 2.7 | 0.65 |
| | Chromafenozide | 200 | 2.423 (1.813–3.329) | 1.506 $\pm$ 0.239 | 2.600 (3) | 2.2 | 0.43 |
| | Pyridalyl | 200 | 10.112 (6.389–13.964) | 1.377 $\pm$ 0.241 | 1.610 (3) | 7.3 | 0.69 |
| ND | Metaflumizone | 200 | 216.677 (171.899–274.787) | 1.997 $\pm$ 0.263 | 2.385 (3) | 50.8 | 0.50 |
| | Chlorantraniliprole | 200 | 186.983 (150.143–232.434) | 2.181 $\pm$ 0.273 | 1.832 (3) | 64.3 | 0.66 |
| | Cyantraniliprole | 200 | 14.312 (11.354–18.617) | 1.966 $\pm$ 0.266 | 2.569 (3) | 8.4 | 0.45 |
| | Tetraniliprole | 200 | 0.604 (0.417–0.829) | 1.394 $\pm$ 0.204 | 1.004 (3) | 4.9 | 0.79 |
| | Chromafenozide | 200 | 7.252 (1.813–3.329) | 2.023 $\pm$ 0.266 | 2.207 (3) | 6.7 | 0.52 |
| | Pyridalyl | 200 | 33.708 (26.335–45.504) | 1.845 $\pm$ 0.261 | 2.880 (3) | 24.2 | 0.40 |

[a] Number of tested larvae. [b] Median lethal concentration and 95% confidence interval. [c] RR: Resistance Ratio = LC$_{50}$ (field-collected population)/LC$_{50}$ (Lab-S).3.4. Monitoring Sensitivity to Six Insecticides in Southern China.

**Table 6.** Insecticide susceptibility of *S. litura* collected from southern China.

| Population | Insecticide | N [a] | LC$_{50}$ (95% CI; mg/L) [b] | Slope $\pm$ SE | X$^2$ (df) | RR [c] | *p* Value |
|---|---|---|---|---|---|---|---|
| GZ | Metaflumizone | 200 | 341.918 (271.818–425.806) | 2.122 $\pm$ 0.271 | 1.317 (3) | 80.2 | 0.73 |
| | Chlorantraniliprole | 200 | 324.233 (248.476–449.835) | 1.671 $\pm$ 0.250 | 1.525 (3) | 111.6 | 0.70 |
| | Cyantraniliprole | 200 | 25.887 (19.194–36.562) | 1.442 $\pm$ 0.235 | 0.905 (3) | 15.2 | 0.80 |
| | Tetraniliprole | 200 | 0.457 (0.326–0.737) | 1.287 $\pm$ 0.235 | 2.306 (3) | 3.7 | 0.51 |
| | Chromafenozide | 200 | 9.940 (7.645–12.940) | 1.740 $\pm$ 0.248 | 2.064 (3) | 9.2 | 0.59 |
| | Pyridalyl | 200 | 51.219 (40.040–65.596) | 1.874 $\pm$ 0.255 | 1.507 (3) | 36.7 | 0.70 |
| SY | Metaflumizone | 200 | 176.440 (142.645–220.878) | 2.204 $\pm$ 0.288 | 2.291 (3) | 41.4 | 0.51 |
| | Chlorantraniliprole | 200 | 221.704 (167.731–288.184) | 1.699 $\pm$ 0.245 | 1.942 (3) | 76.3 | 0.65 |
| | Cyantraniliprole | 200 | 18.026 (14.414–22.436) | 2.150 $\pm$ 0.272 | 1.561 (3) | 10.6 | 0.70 |
| | Tetraniliprole | 200 | 0.254 (0.187–0.324) | 1.960 $\pm$ 0.279 | 1.024 (3) | 2.0 | 0.79 |
| | Chromafenozide | 200 | 3.015 (2.379–3.713) | 2.295 $\pm$ 0.298 | 2.439 (3) | 2.8 | 0.49 |
| | Pyridalyl | 200 | 26.464 (19.966–36.465) | 1.546 $\pm$ 0.241 | 0.938 (3) | 19.0 | 0.80 |
| GL | Metaflumizone | 200 | 27.666 (22.166–34.732) | 2.110 $\pm$ 0.270 | 1.713 (3) | 6.5 | 0.67 |
| | Chlorantraniliprole | 200 | 1.065 (0.707–1.420) | 1.678 $\pm$ 0.264 | 0.436 (3) | 0.4 | 0.93 |
| | Cyantraniliprole | 200 | 1.275 (0.903–1.661) | 1.751 $\pm$ 0.261 | 0.488 (3) | 0.7 | 0.92 |
| | Tetraniliprole | 200 | 0.237 (0.192–0.291) | 2.321 $\pm$ 0.284 | 2.193 (3) | 1.9 | 0.54 |
| | Chromafenozide | 200 | 3.263 (2.701–3.971) | 2.627 $\pm$ 0.314 | 2.387 (3) | 3.0 | 0.50 |
| | Pyridalyl | 200 | 0.790 (0.582–0.978) | 2.766 $\pm$ 0.444 | 1.287 (3) | 0.6 | 0.73 |
| YX | Metaflumizone | 200 | 275.731 (214.701–358.255) | 1.805 $\pm$ 0.251 | 2.407 (3) | 64.7 | 0.48 |
| | Chlorantraniliprole | 200 | 171.250 (126.156–227.052) | 1.558 $\pm$ 0.240 | 2.260 (3) | 58.9 | 0.51 |
| | Cyantraniliprole | 200 | 24.834 (18.380–36.707) | 1.446 $\pm$ 0.240 | 1.817 (3) | 14.6 | 0.66 |
| | Tetraniliprole | 200 | 0.396 (0.317–0.489) | 2.249 $\pm$ 0.282 | 1.933 (3) | 3.2 | 0.65 |
| | Chromafenozide | 200 | 15.393 (11.683–21.537) | 1.589 $\pm$ 0.246 | 1.210 (3) | 14.3 | 0.76 |
| | Pyridalyl | 200 | 85.007 (68.595–104.692) | 2.289 $\pm$ 0.282 | 2.070 (3) | 61.0 | 0.59 |

[a] Number of tested larvae. [b] Median lethal concentration and 95% confidence interval. [c] RR: Resistance Ratio = LC$_{50}$ (field-collected population)/LC$_{50}$ (Lab-S).3.5. Pairwise Correlation between the Values of Log LC$_{50}$ of Different Chemical Agents.

Pairwise correlation coefficients were assessed between the values of log $LC_{50}$ of the tested chemical agent for *S. litura* field-collected populations. Analysis of Pearson's correlation was performed to evaluate cross-resistance among different chemical agents. The data were displayed with normality assumption; otherwise, Spearman rank correlation was conducted. As shown in Table 7, resistance to pyridalyl was significantly correlated with metaflumizone resistance (Pearson's correlation coefficient $r = 0.883$, $p < 0.01$), chlorantraniliprole, ($r = 0.926$, $p < 0.01$), cyantraniliprole, ($r = 0.958$, $p < 0.01$) and chromafenozide ($r = 0.758$, $p < 0.01$). Similarly, there were significant positive correlations between resistance to metaflumizone and chlorantraniliprole ($r = 0.926$, $p < 0.01$), and cyantraniliprole ($r = 0.958$, $p < 0.01$). Moreover, significant correlations were detected between chlorantraniliprole and cyantraniliprole ($r = 0.969$, $p < 0.01$). In contrast, no significant correlations between tetraniliprole and the other five tested chemical agents were detected in field-collected populations of *S. litura* in China ($p > 0.05$).

**Table 7.** Pairwise correlation analysis of the $LC_{50}$ values for six insecticides in the eleven field populations of *S. litura*.

| | Metaflumizone | Chlorantraniliprole | Cyantraniliprole | Tetraniliprole | Chromafenozide |
|---|---|---|---|---|---|
| Chlorantraniliprole | 0.918 ** | | | | |
| Cyantraniliprole | 0.916 ** | 0.969 ** | | | |
| Tetraniliprole | −0.067 | −0.002 | −0.005 | | |
| Chromafenozide | 0.545 | 0.624 * | 0.722 * | 0.344 | |
| Pyridalyl | 0.883 ** | 0.926 ** | 0.958 ** | 0.219 | 0.758 * |

** indicates significant difference ($p < 0.01$); * indicates significant difference ($p < 0.05$).

## 4. Discussion

Monitoring of pesticide resistance is one necessary part of management of resistance, and it has been considered as imperative for the strategies of pest management [23]. In China, previous research has determined the baseline toxicities of insecticides used to control *S. litura* as well as the corresponding susceptibility levels [4,7–9]. Our results indicate that most *S. litura* populations (eight of eleven) displayed significant resistance to metaflumizone, and some *S. litura* populations (four of eleven) also showed significant resistance to chlorantraniliprole. Similarly, four of the eleven field-collected populations presented significant resistance to pyridalyl. Additionally, some *S. litura* populations (three of eleven) showed moderate resistance to cyantraniliprole. In China, the levels of resistance to metaflumizone and chlorantraniliprole in *S. litura* were monitored, and high to very high levels of resistance to them were detected, respectively [7,12], and cases of low to moderate resistance to cyantraniliprole were reported previously [9].

In contrast to the resistance statuses of the above four tested insecticides, regarding chromafenozide resistance in the eleven field-collected *S. litura* populations, we found that one population showed moderate resistance, and two populations showed low resistance. Furthermore, we found that only one of the eleven tested populations showed low resistance to tetraniliprole, and all other populations were susceptible to this chemical agent. In China, although field-evolved resistance to tetraniliprole has been reported in *Spodoptera exigua* [24], few cases of resistance in *S. litura* were reported until now. Similarly, to date in China, there are no reports of resistance to chromafenozide in lepidopteran pests. Considering that application of pesticides is still one of main measures for pest management [25], our above results indicate that the novel anthranilic diamide insecticide tetraniliprole and the dibenzoylhydrazine insecticide chromafenozide are still effective chemistries that farmers can use to control *S. litura* in China.

To avoid or slow the development of pesticide resistance, the study on cross-resistance between pesticides could guide the rotation and mixed application of them [26]. Previous publications have indicated that cross-resistance was detected among anthranilic diamide pesticides, such as metaflumizone and pyridalyl, in lepidopteran pests [9,12,19,27]. In the current study, pairwise correlation of values of log $LC_{50}$ also found the existence of cross-

resistance among long-term use insecticides, including chlorantraniliprole, cyantraniliprole, metaflumizone, pyridalyl, and chromafenozide. In contrast, tetraniliprole presented little cross-resistance with tested insecticides in the 11 field-collected populations of *S. litura* in China. Similarly, Indian field populations of *Spodoptera frugiperda* displayed little cross-resistance to all the tested pesticides such as organophosphates, carbamates, pyrethroids, fiproles, avermectins, spinosyns and anthranilic diamides [20]. The lack of cross-resistance to tetraniliprole makes it a promising method for the management of resistance in the field. Besides, a robust approach to resistance management is expected to take into account of rotation of tetraniliprole with other chemical classes of pesticides, such as pyridalyl and chromafenozide, so as to extend the life span of tetraniliprole in the field of China. Additionally, alternative strategies to control insect pests for example biological control of pests will be important while they develop insecticide resistance [28,29]. Recently, *Autographa californica Multiple Nucleopolyhedrovirus* (AcMNPV), which is a baculovirus that causes systemic infections in many arthropod pests, was indicated that it could be promising for the management of *S. frugiperda* and *Trichoplusia ni* [30].

**Author Contributions:** Conceptualization, W.L. and R.W.; methodology, Z.Z. and B.G.; software, Z.Z. and B.G.; validation, C.Q.; formal analysis, Z.Z. and B.G.; investigation, Z.Z. and B.G.; resources, J.G.; data curation, C.Q.; writing—original draft preparation, W.L. and R.W.; writing—review and editing, W.L. and R.W.; visualization, R.W.; supervision, R.W.; project administration, C.L. and R.W; funding acquisition, C.L. and R.W. All authors have read and agreed to the published version of the manuscript.

**Funding:** This research was supported by the China Agriculture Research System of MOF and MARA, the Scientific and Technological Innovation Capacity Construction Special Funds of the Beijing Academy of Agriculture and Forestry Sciences, Beijing, China (KJCX20210437).

**Institutional Review Board Statement:** Not applicable.

**Informed Consent Statement:** Not applicable.

**Data Availability Statement:** Not applicable.

**Acknowledgments:** We acknowledge the excellent technical assistance and collection of field populations from Caihua Shi, Yong Fang, and Jinda Wang.

**Conflicts of Interest:** The authors declare no conflict of interest.

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
