# Peer review of "Resistance Monitoring for Six Insecticides in Vegetable Field-Collected Populations of Spodoptera litura from China"

_horticulturae, doi:10.3390/horticulturae8030255_

Round 1

Reviewer 1 Report

The manuscript submitted by Zhang et al, titled "Resistance Monitoring for Six Insecticides in Vegetable Field Collected Populations of Spodoptera litura from China” evaluated the effect of six insecticides in field-collected populations of S. litura. Overall, the manuscript is well written. The method section needs to be elaborated. Further, if the insects are developing resistance to insecticides other strategies need to be discussed and suggested in the discussion section. I have a few comments that will improve the readability of this paper.

Major comments

What does the color pellet in figure 2 represent?

The discussion needs to include the section for alternative strategies to control insect pests while they develop insecticide-resistant for example biological control of insects. Recently, Pantha et al., 2021 have identified the possibility of using Autographa californica Multiple Nucleopolyhedrovirus (AcMNPV) since several of the host genes were targeted by AcMNPV are also targets of chemical insecticides currently used to control arthropod pests.

I suggest the author cite the following papers in the discussion section.

Reference:

1. Pantha P, Chalivendra S, Oh DH, Elderd BD, Dassanayake M (2021) A tale of two transcriptomic responses in agricultural pests via host defenses and viral replication. Int J Mol Sci 22: 1–27.

2. Cory JS, Hirst ML, Williams T, Hails RS, Goulson D, Green BM, Carty TM, Possee RD, Cayley PJ, Bishop DHL (1994) Field trial of a genetically improved baculovirus insecticide. Nature 370: 138–140

3. Shim HJ, Choi JY, Wang Y, Tao XY, Liu Q, Roh JY, Kim JS, Kim WJ, Woo SD, Jin BR, et al (2013) Neurobactrus, a novel, highly effective, and environmentally friendly recombinant baculovirus insecticide. Appl Environ Microbiol 79: 141–149

Author Response

Please check the point-by-point response to the reviewer’s comments attached, thanks a lot.

Reviewer 2 Report

The study has been carried out to monitor the resistance development of S. litura against some widely used insecticides. This is an important piece of information. I found the study is well designed. However, some issues, as suggested below, are required to be addressed well in order to improve the manuscript. Please find below:

Line 33: Please write ‘causes’ for inflicts.

Line 49: please write ‘pesticide chemistry’, not pesticide chemistries.

Line 93: Please write ‘minor modification’ instead of minor revision.

2.3. Bioassays: Why did researchers select third-instar larvae for bioassay? Please mention briefly.

Line 116: Please delete ‘3.1.’ from the beginning of the sentence.

Unit: Please check the better way of writing mg/L or mg a.i./L.

Table 3 & 4: Please check the baseline susceptibility for Lab-S strain and values for the population collected from WH. This is exactly the same as for the population collected from WH! Please check.

Line 184-188: ‘In the current study…………cross-resistance among them’.Please do not repeat it.

Line 189: Eight of eleven displayed significant resistance to Metaflumizone. Please check the values for Lab-S and WH populations. Otherwise, it is not clear. The same is for other insecticides.

The discussion part should include the presumable reason for this development of resistance and briefly the possible way to mitigate it.  

Author Response

(The authors gave the same response as above.)

Reviewer 3 Report

In the manuscript (ms) entitled “Resistance Monitoring for Six Insecticides in Vegetable Field- Collected Populations of Spodoptera litura from China”. Authors received fascinating results of field management programs aimed at controlling S. litura, and were able to logically interpret their data. This Reviewer has some comments that should be addressed to improve the presentation and readability of the ms:

1. The abstract reflects what the authors found in this study. Author may include a sentence at the end of the summary (conclusion) whether this study would provide any useful advice to the readers?

2. In the introduction, the application of insecticides and their mode of action against cutworm is completely missing, Please add this information.

3. Materials and Methods: it is not clear that the field experiment were done in replicates?

4. Discussion: This section seems more descriptive; authors should compile their results with the published papers for easy understanding. In addition, authors should highlight some examples and discuss.

5. Figure legends should be provided in detail (meaning self explanatory).

6. Tables must have significance letters to indicate the level of P value.

Author Response

(The authors gave the same response as above.)

Round 2

Reviewer 2 Report

The manuscript has been improved. Thank you.